# A Primal Approach to Constrained Policy Optimization: Global Optimality and Finite-Time Analysis

## Abstract

Safe reinforcement learning (SRL) problems are typically modeled as constrained Markov Decision Process (CMDP), in which an agent explores the environment to maximize the expected total reward and meanwhile avoids violating certain constraints on a number of expected total costs. In general, such SRL problems have nonconvex objective functions subject to multiple nonconvex constraints, and hence are very challenging to solve, particularly to provide a globally optimal policy. Many popular SRL algorithms adopt a primal-dual structure which utilizes the updating of dual variables for satisfying the constraints. In contrast, we propose a primal approach, called constraint-rectified policy optimization (CRPO), which updates the policy alternatingly between objective improvement and constraint satisfaction. CRPO provides a primal-type algorithmic framework to solve SRL problems, where each policy update can take any variant of policy optimization step. To demonstrate the theoretical performance of CRPO, we adopt natural policy gradient (NPG) for each policy update step and show that CRPO achieves an $\mathcal{O}(1/\sqrt{T})$ convergence rate to the global optimal policy in the constrained policy set and an $\mathcal{O}(1/\sqrt{T})$ error bound on constraint satisfaction. This is the first finite-time analysis of SRL algorithms with global optimality guarantee. Our empirical results demonstrate that CRPO can outperform the existing primal-dual baseline algorithms significantly.

## 1 Introduction

Reinforcement learning (RL) has achieved great success in solving complex sequential decision-making and control problems such as Go Silver et al. (2017), StarCraft DeepMind (2019) and recommendation system Zheng et al. (2018), etc. In these settings, the agent is allowed to explore the entire state and action space to maximize the expected total reward. However, in safe RL, in addition to maximizing the reward, an agent needs to satisfy certain constraints. Examples include self-driving cars Fisac et al. (2018), cellular network Julian et al. (2002), and robot control Levine et al. (2016). One standard model for safe RL is constrained Markov Decision Process (CMDP) Altman (1999), which further requires the policy to satisfy the constraints on a number of accumulated costs. The global optimal policy in this setting is the one that maximizes the reward and at the same time satisfies the cost constraints. In general, it is very challenging to find the global optimal policy in CMDP, as both the objective and constraints are nonconvex functions.

A commonly used approach to solve CMDP is the *primal-dual* method Chow et al. (2017); Tessler et al. (2018); Ding et al. (2020a); Stooke et al. (2020), in which the constrained problem is converted to an unconstrained one by augmenting the objective with a sum of constraints weighted by their corresponding Lagrange multipliers. Usually, Lagrange multipliers are updated in the dual space concurrently Tessler et al. (2018). Although it has been observed that primal-dual methods always converge to the feasible set in the end Ray et al. (2019), such an approach is sensitive to the initialization of Lagrange multipliers and the learning rate, thus can incur extensive cost in hyperparameter tuning Achiam et al. (2017); Chow et al. (2019). Another baseline approach is *constrained policy optimization (CPO)*, in which a linearized constrained problem is solved from scratch at each iteration to obtain the policy in the next step. However, a successful implementation of CPO requires a feasible initialization, which by itself can be very difficult especially with multiple constraints Ray et al.

(2019). Other approaches such as Lyapunov method Chow et al. (2018; 2019), safety layer method Dalal et al. (2018a) and interior point method Liu et al. (2019b) have also been proposed recently. However, these methods do not have clear guidance in hyperparameter tuning, and thus suffer from nontrivial cost to implement in practice Stooke et al. (2020).

**Thus, one goal here is to design an easy-to-implement SRL algorithm that enjoys the ease as uncontrained problems and readily approaches feasible points from random initialization.**

In contrast to the extensive empirical studies of SRL algorithms, theoretical understanding of the convergence properties of SRL algorithms is very limited. Tessler et al. (2018) provided an asymptotic convergence analysis for primal-dual method and established a local convergence guarantee under certain stability assumptions. Paternain et al. (2019) showed that the primal-dual method achieves zero duality gap, which can imply the global optimality under certain assumptions. Recently, Ding et al. (2020a) proposed a primal-dual type proximal policy optimization (PPO) and established the regret bound for linear CMDP. The convergence rate of primal-dual method is characterized in a concurrent work Ding et al. (2020b). So far, there exist no primal-type SRL algorithms that have been shown to enjoy global optimality guarantee under general CMDP. Further, the finite-time performance (convergence rate) has not been characterized for any primal-type SRL algorithm.

**Thus, the second goal here is to establish global optimality guarantee and the finite-time convergence rate for the proposed algorithm under general CMDP.**

## 1.1 MAIN CONTRIBUTIONS

We propose a novel **Constraint-Rectified Policy Optimization (CRPO)** approach for CMDP, where all updates are taken in the primal domain. CRPO applies *unconstrained* policy maximization update w.r.t. the reward on the one hand, and if any constraint is violated, momentarily rectifies the policy back to the feasible set along the descent direction of the violated constraint also by applying *unconstrained* policy minimization update w.r.t. the constraint function. Hence, CRPO can be implemented as easy as unconstrained policy optimization algorithms. It does not introduce heavy hyperparameter tuning to enforce constraint satisfaction, nor does it require initialization to be feasible. CRPO provides a primal-type framework for solving SRL problems, and its optimization update can adopt various well-developed unconstrained policy optimization methods such as natural policy gradient (NPG) Kakade (2002), trust region policy optimization (TRPO) Schulman et al. (2015), PPO, etc.

To provide the theoretical guarantee for CRPO, we adopt NPG as a representative optimizer and investigate the convergence of CRPO in two settings: tabular and function approximation, where in the function approximation setting the state space can be infinite. For both settings, we show that CRPO converges to a global optimum at a convergence rate of $\mathcal{O}(1/\sqrt{T})$. Furthermore, the constraint satisfaction error converges to zero at a rate of $\mathcal{O}(1/\sqrt{T})$. To the best of our knowledge, CRPO is the first primal-type SRL algorithm that has provably global optimality guarantee. This work also provides the first finite-time analysis for SRL algorithm without restrictive assumptions on CMDP.

Our experiments demonstrate that CRPO outperforms the baseline primal-dual algorithm with higher return reward and smaller constraint satisfaction error.

## 1.2 RELATED WORK

**Safe RL and CMDP:** Algorithms based on primal-dual methods have been widely adopted for solving constrained RL problems, such as PDO Chow et al. (2017), RCPO Tessler et al. (2018), OPDOP Ding et al. (2020a) and CPPO Stooke et al. (2020). The effectiveness of primal-dual methods is justified in Paternain et al. (2019), in which zero duality gap is guaranteed under certain assumptions. Constrained policy optimization (CPO) Achiam et al. (2017) extends TRPO to handle constraints, and is later modified with a two-step projection method Yang et al. (2019a). Other methods have also been proposed. For example, Chow et al. (2018; 2019) leveraged Lyapunov functions to handle constraints. Yu et al. (2019) proposed a constrained policy gradient algorithm with convergence guarantee by solving a sequence of sub-problems. Dalal et al. (2018a) proposed to add a safety layer to the policy network so that constraints can be satisfied at each state. Liu et al. (2019b) developed an interior point method for safe RL, which augments the objective with logarithmic barrier functions. This paper proposes a CRPO algorithm, which can be implemented as easy as unconstrained policy optimization methods and has global optimality guarantee under general CMDP.

**Finite-Time Analysis of Policy Optimization:** The finite-time analysis of various policy optimization algorithms have been well studied. The convergence rate of policy gradient (PG) and actor-critic (AC) algorithms have been established in Shen et al. (2019); Papini et al. (2017; 2018); Xu et al. (2020a; 2019); Xiong et al. (2020); Zhang et al. (2019) and Xu et al. (2020b); Wang et al. (2019); Yang et al. (2019b); Kumar et al. (2019); Qiu et al. (2019), respectively, in which PG or AC algorithm is shown to converge to a local optimal. In some special settings such as tabular and LQR, PG and AC can be shown to convergence to the global optimal Agarwal et al. (2019); Yang et al. (2019b); Fazel et al. (2018); Malik et al. (2018); Tu & Recht (2018); Bhandari & Russo (2019; 2020). Algorithms such as NPG, NAC, TRPO and PPO explore the second order information, and achieve great success in practice. These algorithms have been shown to converge to a global optimum in various settings, where the convergence rate has been established in Agarwal et al. (2019); Shani et al. (2019); Liu et al. (2019a); Wang et al. (2019); Cen et al. (2020); Xu et al. (2020c). However, all the above studies only consider unconstrained MDP. A concurrent and independent work Ding et al. (2020b) established the global convergence rate of primal-dual method for CMDP under weak Slater's condition assumption. So far the finite-time performance of primal-type policy optimization in general CMDP settings has not been studied. Our work is the first one that establishes such a result.

## 2 PROBLEM FORMULATION AND PRELIMINARIES

### 2.1 MARKOV DECISION PROCESS

A discounted Markov decision process (MDP) is a tuple $(\mathcal{S}, \mathcal{A}, c_0, \mathsf{P}, \xi, \gamma)$, where $\mathcal{S}$ and $\mathcal{A}$ are state and action spaces; $c_0 : \mathcal{S} \times \mathcal{A} \times \mathcal{S} \to \mathbb{R}$ is the reward function; $\mathsf{P} : \mathcal{S} \times \mathcal{A} \times \mathcal{S} \to [0, 1]$ is the transition kernel, with $\mathsf{P}(s'|s, a)$ denoting the probability of transitioning to state $s'$ from previous state $s$ given action $a$; $\xi : \mathcal{S} \to [0, 1]$ is the initial state distribution; and $\gamma \in (0, 1)$ is the discount factor. A policy $\pi : \mathcal{S} \to \mathcal{P}(\mathcal{A})$ is a mapping from the state space to the space of probability distributions over the actions, with $\pi(\cdot|s)$ denoting the probability of selecting action $a$ in state $s$. When the associated Markov chain $\mathsf{P}(s'|s) = \sum_{\mathcal{A}} P(s'|s, a)\pi(a|s)$ is ergodic, we denote $\mu_\pi$ as the stationary distribution of this MDP, i.e. $\int_{\mathcal{S}} \mathsf{P}(s'|s)\mu_\pi(ds) = \mu_\pi(s')$. Moreover, we define the visitation measure induced by the police $\pi$ as $\nu_\pi(s, a) = (1 - \gamma) \sum_{t=0}^{\infty} \gamma^t \mathsf{P}(s_t = s, a_t = a)$.

For a given policy $\pi$, we define the state value function as $V_\pi^0(s) = \mathbb{E}[\sum_{t=0}^{\infty} \gamma^t c_0(s_t, a_t, s_{t+1})|s_0 = s, \pi]$, the state-action value function as $Q_\pi^0(s, a) = \mathbb{E}[\sum_{t=0}^{\infty} \gamma^t c_0(s_t, a_t, s_{t+1})|s_0 = s, a_0 = a, \pi]$, and the advantage function as $A_\pi^0(s, a) = Q_\pi^0(s, a) - V_\pi^0(s)$. In reinforcement learning, we aim to find an optimal policy that maximizes the expected total reward function defined as $J_0(\pi) = \mathbb{E}[\sum_{t=0}^{\infty} \gamma^t c_0(s_t, a_t, s_{t+1})] = \mathbb{E}_\xi[V_\pi^0(s)] = \mathbb{E}_{\xi \cdot \pi}[Q_\pi^0(s, a)]$.

### 2.2 CONSTRAINED MARKOV DECISION PROCESS

A constrained Markov Decision Process (CMDP) is an MDP with additional constraints that restrict the set of allowable policies. Specifically, when taking action at some state, the agent can incur a number of costs denoted by $c_1, \cdots, c_p$, where each cost function $c_i : \mathcal{S} \times \mathcal{A} \times \mathcal{S} \to \mathbb{R}$ maps a tuple $(s, a, s')$ to a cost value. Let $J_i(\pi)$ denotes the expected total cost function with respect to $c_i$ as $J_i(\pi) = \mathbb{E}[\sum_{t=0}^{\infty} \gamma^t c_i(s_t, a_t, s_{t+1})]$. The goal of the agent in CMDP is to solve the following constrained problem

$$\max_\pi J_0(\pi), \quad \text{subject to } J_i(\pi) \leq d_i, \quad \forall i = 1, \cdots, p, \tag{1}$$

where $d_i$ is a fixed limit for the $i$-th constraint. We denote the set of feasible policies as $\Omega_C \equiv \{\pi : \forall i, J_i(\pi) \leq d_i\}$, and define the optimal policy for CMDP as $\pi^* = \arg\min_{\pi \in \Omega_C} J_0(\pi)$. For each cost $c_i$, we define its corresponding state value function $V_\pi^i$, state-action value function $Q_\pi^i$, and advantage function $A_\pi^i$ analogously to $V_\pi^0, Q_\pi^0$, and $A_\pi^0$, with $c_i$ replacing $c_0$, respectively.

### 2.3 POLICY PARAMETERIZATION AND POLICY GRADIENT

In practice, a convenient way to solve the problem eq. (1) is to parameterize the policy and then optimize the policy over the parameter space. Let $\{\pi_w : \mathcal{S} \to \mathcal{P}(\mathcal{A})|w \in \mathcal{W}\}$ be a parameterized policy class, where $\mathcal{W}$ is the parameter space. Then, the problem in eq. (1) becomes

$$\max_{w \in \mathcal{W}} J_0(\pi_w), \quad \text{subject to } J_i(\pi_w) \leq d_i, \quad \forall i = 1, \cdots, p, \tag{2}$$

The policy gradient of the function $J_i(\pi_w)$ has been derived by Sutton et al. (2000) as $\nabla J_i(\pi_w) = \mathbb{E}[Q_{\pi_w}^i(s, a)\phi_w(s, a)]$, where $\phi_w(s, a) := \nabla_w \log \pi_w(a|s)$ is the score function. Furthermore, the

---

**Algorithm 1** Constraint-Rectified Policy Optimization (CRPO)

---
1: **Initialize:** initial parameter $w_0$, empty set $\mathcal{N}_0$
2: **for** $t = 0, \cdots, T - 1$ **do**
3:     Policy evaluation under $\pi_{w_t}$ to obtain $\bar{Q}_t^i(s, a) \approx Q_{\pi_{w_t}}^i(s, a)$
4:     Sample $(s_j, a_j) \in \mathcal{B}_t$, compute $\bar{J}_{i,\mathcal{B}_t} = \sum_{j \in \mathcal{B}_t} \rho_{j,t} \bar{Q}_t^i(s_j, a_j)$, for $i = 0, \cdots, p$, ($\rho_{j,t}$ is the weight)
5:     **if** $\bar{J}_{i,\mathcal{B}_t} \leq d_i + \eta$ for all $i = 1, \cdots, p$, **then**
6:         Add $w_t$ into set $\mathcal{N}_0$
7:         Take one-step policy update towards maximize $J_0(w_t)$: $w_t \to w_{t+1}$
8:     **else**
9:         Choose any $i_t \in \{1, \cdots, p\}$ such that $\bar{J}_{i_t, \mathcal{B}_t} > d_{i_t} + \eta$
10:         Take one-step policy update towards minimize $J_{i_t}(w_t)$: $w_t \to w_{t+1}$
11:     **end if**
12: **end for**
13: **Output:** $w_{\text{out}}$ randomly chosen from $\mathcal{N}_0$ with uniform distribution

---

natural policy gradient was defined by Kakade (2002) as $\Delta_i(w) = F(w)^\dagger \nabla J_i(\pi_w)$, where $F(w)$ is the Fisher information matrix defined as $F(w) = \mathbb{E}_{\nu_{\pi_w}}[\phi_w(s, a)\phi_w(s, a)^\top]$.

## 3 Constraint-Rectified Policy Optimization (CRPO) Algorithm

In this section, we propose the CRPO (see Algorithm 1) approach for solving CMDP problem in eq. (2). The idea of CRPO lies in updating the policy to maximize the unconstrained objective function $J_0(\pi_{w_t})$ of the reward, alternatingly with rectifying the policy to reduce a constraint function $J_i(\pi_{w_t})$ $(i \geq 1)$ (along the descent direction of this constraint) if it is violated. Each iteration of CRPO consists of the following three steps.

**Policy Evaluation:** At the beginning of each iteration, we estimate the state-action value function $\bar{Q}_{\pi_t}^i(s, a) \approx Q_{\pi_{w_t}}^i(s, a)$ $(i = \{0, \cdots, p\})$ for both reward and costs under current policy $\pi_{w_t}$.

**Constraint Estimation:** After obtaining $\bar{Q}_{\pi_t}^i$, the constraint function $J_i(w_t) = \mathbb{E}_{\xi \cdot \pi_{w_t}}[Q_{w_t}^i(s, a)]$ can then be approximated via a weighted sum of approximated state-action value function: $\bar{J}_{i,\mathcal{B}_t} = \sum_{j \in \mathcal{B}_t} \rho_{j,t} \bar{Q}_t^i(s_j, a_j)$. Note this step does not take additional sampling cost, as the generation of samples $(s_j, a_j) \in \mathcal{B}_t$ does not require the agent to interact with the environment.

**Policy Optimization:** We then check whether there exists an $i_t \in \{1, \cdots, p\}$ such that the approximated constraint $\bar{J}_{i_t,\mathcal{B}_t}$ violates the condition $\bar{J}_{i_t,\mathcal{B}_t} \leq d_i + \eta$, where $\eta$ is the tolerance. If so, we take **one-step** update of the policy towards minimizing the corresponding constraint function $J_{i_t}(\pi_{w_t})$ to enforce the constraint. If multiple constraints are violated, we can choose to minimize any one of them. If all constraints are satisfied, we take **one-step** update of the policy towards maximizing the objective function $J_0(\pi_{w_t})$. To apply CRPO in practice, we can use any policy optimization update such as NPG, TRPO, PPO Schulman et al. (2017), ACKTR Wu et al. (2017), and SAC Haarnoja et al. (2018), etc, in the policy optimization step (line 7 and line 10).

Differently from previous SRL algorithms, which usually take nontrivial costs to deal with the constraints Chow et al. (2017); Tessler et al. (2018); Yang et al. (2019a); Chow et al. (2018; 2019); Liu et al. (2019b); Dalal et al. (2018a), our CRPO algorithm essentially performs unconstrained policy optimization alternatingly on different objectives during the training, and thus can be implemented as easy as unconstrained policy optimization algorithms without introducing heave hyperparameter tuning and additional initialization requirement.

CRPO algorithm is inspired by, yet very different from the cooperative stochastic approximation (CSA) method Lan & Zhou (2016) in optimization literature. First, CSA is designed for convex optimization subject to convex functional constraint, and is thus not capable of handling the more challenging SRL problems eq. (2), which are nonconvex optimization subject to nonconvex functional constraints. Second, CSA is designed to handle only a single constraint, whereas CRPO can handle multiple constraints with guaranteed constraint satisfaction and global optimality. Third, CSA assumes the accessibility of unbiased estimators of both gradient and constraint, while in our problem both the NPG update and constraints are estimated through the random output from the critic, thus requiring developing a new analysis framework to handle this more challenging setting.

## 4 CONVERGENCE ANALYSIS OF CRPO

In this section, we take NPG as a representative optimizer in CRPO, and establish the global convergence rate of CRPO in both the tabular and function approximation settings. Note that TRPO and ACKTR update can be viewed as the NPG approach with adaptive stepsize. Thus, the global convergence property we establish for NPG implies similar convergence guarantee of CRPO that takes TRPO or ACKTR as the optimizer.

### 4.1 TABULAR SETTING

In the tabular setting, we consider the softmax parameterization. For any $w \in \mathbb{R}^{|\mathcal{S}| \times |\mathcal{A}|}$, the corresponding softmax policy $\pi_w$ is defined as

$$\pi_w(a|s) := \frac{\exp(w(s,a))}{\sum_{a' \in \mathcal{A}} \exp(w(s,a'))}, \quad \forall (s,a) \in \mathcal{S} \times \mathcal{A}. \tag{3}$$

Clearly, the policy class defined in eq. (3) is complete, as any stochastic policy in the tabular setting can be represented in this class.

**Policy Evaluation:** To perform the policy evaluation in Algorithm 1 (line 3), we adopt the temporal difference (TD) learning, in which a vector $\theta^i \in \mathbb{R}^{|\mathcal{S}| \times |\mathcal{A}|}$ is used to estimate the state-action value function $Q^i_{\pi_w}$ for all $i = 0, \cdots, p$. Specifically, each iteration of TD learning takes the form of

$$\theta^i_{k+1}(s,a) = \theta^i_k(s,a) + \beta_k[c_i(s,a,s') + \gamma \theta^i_k(s',a') - \theta^i_k(s,a)], \tag{4}$$

where $s \sim \mu_{\pi_w}$, $a \sim \pi_w(\cdot|s)$, $s' \sim \mathsf{P}(\cdot|s,a)$, $a' \sim \pi_w(\cdot|s')$, and $\beta_k$ is the learning rate. In line 3 of Algorithm 1, we perform the TD update in eq. (4) for $K_{\text{in}}$ iterations. It has been shown in Dalal et al. (2018b) that the iteration in eq. (4) of TD learning converges to a fixed point $\theta^i_*(\pi_w) \in \mathbb{R}^{|\mathcal{S}| \times |\mathcal{A}|}$. Each component of the fixed point is the corresponding state-action value: $\theta^i_*(\pi_w)(s,a) = Q^i_{\pi_w}(s,a)$. The following lemma characterizes the convergence rate of TD learning in the tabular setting.

**Lemma 1** (Dalal et al. (2019)). *Consider the iteration given in eq. (4) with arbitrary initialization $\theta^i_0$. Assume that the stationary distribution $\mu_{\pi_w}$ is not degenerate for all $w \in \mathbb{R}^{|\mathcal{S}| \times |\mathcal{A}|}$. Let stepsize $\beta_k = \Theta(\frac{1}{t^\sigma})$ $(0 < \sigma < 1)$. Then, with probability at least $1 - \delta$, we have*

$$\left\| \theta^i_K - \theta^i_*(\pi_w) \right\|_2 = \mathcal{O}\left( \frac{\log(|\mathcal{S}|^2 |\mathcal{A}|^2 K^2/\delta)}{(1-\gamma)K^{\sigma/2}} \right).$$

Note that $\sigma$ can be arbitrarily close to 1. After performing $K_{\text{in}}$ iterations of TD learning as eq. (4), we let $\bar{Q}^i_t(s,a) = \theta^i_{K_{\text{in}}}(s,a)$ for all $(s,a) \in \mathcal{S} \times \mathcal{A}$ and all $i = \{0, \cdots, p\}$. Lemma 1 implies that we can obtain an approximation $\bar{Q}^i_t$ such that $\left\| \bar{Q}^i_t - Q^i_{\pi_w} \right\|_2 = \mathcal{O}(1/\sqrt{K_{\text{in}}})$ with high probability.

**Constraint Estimation:** In the tabular setting, we let the sample set $\mathcal{B}_t$ include all state-action pairs, i.e., $\mathcal{B}_t = \mathcal{S} \times \mathcal{A}$, and the weight factor be $\rho_{j,t} = \xi(s_j)\pi_{w_t}(a_j|s_j)$ for all $t = 0, \cdots, T-1$. Then, the estimation error of the constraints can be upper bounded as $|\bar{J}_i(\theta^i_t) - J_i(w_t)| = |\mathbb{E}[\bar{Q}^i_t(s,a)] - \mathbb{E}[Q^i_{\pi_{w_t}}(s,a)]| \leq ||\bar{Q}^i(\theta^i_t) - Q^i_{\pi_w}||^2$. Thus, our approximation of constraints is accurate when the approximated value function $\bar{Q}^i_t(s,a)$ is accurate.

**Policy Optimization:** In the tabular setting, the natural policy gradient of $J_i(\pi_w)$ is derived by Agarwal et al. (2019) as $\Delta_i(w)_{s,a} = (1-\gamma)^{-1}Q^i_{\pi_w}(s,a)$. Once we obtain an approximation $\bar{Q}^i_t(s,a) \approx Q^i_{\pi_w}(s,a)$, we can use it to update the policy in the upcoming policy optimization step:

$$w_{t+1} = w_t + \alpha \bar{\Delta}_t, \text{ (line 7)} \qquad \text{or} \quad w_{t+1} = w_t - \alpha \bar{\Delta}_t \text{ (line 10)}, \tag{5}$$

where $\alpha > 0$ is stepsize and $\bar{\Delta}_t(s,a) = (1-\gamma)^{-1}\bar{Q}^0_t(s,a)$ (line 7) or $(1-\gamma)^{-1}\bar{Q}^{i_t}_t(s,a)$ (line 10). Recall that $\pi^*$ denotes the optimal policy in the feasible set $\Omega_C$. The following theorem characterizes the convergence rate of CRPO in terms of the objective function and constraint error bound.

**Theorem 1.** *Consider Algorithm 1 in the tabular setting with softmax policy parameterization defined in eq. (3) and any initialization $w_0 \in \mathbb{R}^{|\mathcal{S}| \times |\mathcal{A}|}$. Suppose that the policy evaluation update in eq. (4) takes $K_{in} = \Theta(T^{1/\sigma}(1-\gamma)^{-2/\sigma}\log^{2/\sigma}(T^{1+2/\sigma}/\delta))$ iterations. Let the tolerance $\eta = \Theta(\sqrt{|\mathcal{S}||\mathcal{A}|}/((1-\gamma)^{1.5}\sqrt{T}))$ and perform the NPG update defined in eq. (5) with $\alpha = (1-\gamma)^{1.5}/\sqrt{|\mathcal{S}||\mathcal{A}|T}$. Then, with probability at least $1 - \delta$, we have*

$$J_0(\pi^*) - \mathbb{E}[J_0(w_{out})] \leq \Theta\left( \frac{\sqrt{|\mathcal{S}||\mathcal{A}|}}{(1-\gamma)^{1.5}\sqrt{T}} \right) \quad \text{and} \quad \mathbb{E}[J_i(w_{out})] - d_i \leq \Theta\left( \frac{\sqrt{|\mathcal{S}||\mathcal{A}|}}{(1-\gamma)^{1.5}\sqrt{T}} \right)$$

*for all $i = \{1, \cdots, p\}$, where the expectation is taken with respect to selecting $w_{out}$ from $\mathcal{N}_0$.*

As shown in Theorem 1, starting from an arbitrary initialization, CRPO algorithm is guaranteed to converge to the globally optimal policy $\pi^*$ in the feasible set $\Omega_C$ at a sublinear rate $\mathcal{O}(1/\sqrt{T})$, and the constraint satisfaction error of the output policy converges to zero also at a sublinear rate $\mathcal{O}(1/\sqrt{T})$. Thus, to attain a $w_{out}$ that satisfies $J_0(\pi^*) - \mathbb{E}[J_0(w_{out})] \leq \epsilon$ and $\mathbb{E}[J_i(w_{out})] - d_i \leq \epsilon$ for all $1 \leq i \leq p$, CRPO needs at most $T = \mathcal{O}(\epsilon^{-2})$ iterations, with each policy evaluation step consists of approximately $K_{in} = \mathcal{O}(T)$ iterations when $\sigma$ is close to 1.

## 4.2 FUNCTION APPROXIMATION SETTING

In the function approximation setting, we parameterize the policy by a two-layer neural network together with the softmax policy. We assign a feature vector $\psi(s, a) \in \mathbb{R}^d$ with $d \geq 2$ for each state-action pair $(s, a)$. Without loss of generality, we assume that $\|\psi(s, a)\|_2 \leq 1$ for all $(s, a) \in \mathcal{S} \times \mathcal{A}$. A two-layer neural network $f((s, a); W, b)$ with input $\psi(s, a)$ and width $m$ takes the form of

$$f((s, a); W, b) = \frac{1}{\sqrt{m}} \sum_{r=1}^{m} b_r \cdot \text{ReLU}(W_r^\top \psi(s, a)), \quad \forall(s, a) \in \mathcal{S} \times \mathcal{A}, \tag{6}$$

where $\text{ReLU}(x) = \mathbb{1}(x > 0) \cdot x$, and $b = [b_1, \cdots, b_m]^\top \in \mathbb{R}^m$ and $W = [W_1^\top, \cdots, W_m^\top]^\top \in \mathbb{R}^{md}$ are the parameters. When training the two-layer neural network, we initialize the parameter via $[W_0]_r \sim D_w$ and $b_r \sim \text{Unif}[-1, 1]$ independently, where $D_w$ is a distribution that satisfies $d_1 \leq \|[W_0]_r\|_2 \leq d_2$ (where $d_1$ and $d_2$ are positive constants), for all $[W_0]_r$ in the support of $D_w$. During training, we only update $W$ and keep $b$ fixed, which is widely adopted in the convergence analysis of neural networks Cai et al. (2019); Du et al. (2018). For notational simplicity, we write $f((s, a); W, b)$ as $f((s, a); W)$ in the sequel. Using the neural network in eq. (6), we define the softmax policy

$$\pi_W^\tau(a|s) := \frac{\exp(\tau \cdot f((s, a); W))}{\sum_{a' \mathcal{A}} \exp(\tau \cdot f((s, a'); W))}, \quad \forall(s, a) \in \mathcal{S} \times \mathcal{A}, \tag{7}$$

where $\tau$ is the temperature parameter, and it can be verified that $\pi_W^\tau(a|s) = \pi_{\tau W}(a|s)$. We define the feature mapping $\phi_W(s, a) = [\phi_W^1(s, a)^\top, \cdots, \phi_W^m(s, a)^\top]^\top: \mathbb{R}^d \to \mathbb{R}^{md}$ as

$$\phi_W^r(s, a)^\top = \frac{b_r}{\sqrt{m}} \mathbb{1}(W_r^\top \psi(s, a) > 0) \cdot \psi(s, a), \forall(s, a) \in \mathcal{S} \times \mathcal{A}, \forall r \in \{1, \cdots, m\}.$$

**Policy Evaluation:** To estimate the state-action value function in Algorithm 1 (line 3), we adopt another neural network $f((s, a); \theta^i)$ as an approximator, where $f((s, a); \theta^i)$ has the same structure as $f((s, a); W)$, with $W$ replaced by $\theta \in \mathbb{R}^{md}$ in eq. (7). To perform the policy evaluation step, we adopt the neural TD method proposed in Cai et al. (2019). Specifically, we choose the same initialization as the policy neural work, i.e., $\theta_0^i = W_0$, and perform the neural TD iteration as

$$\theta_{k+1/2}^i = \theta_k^i + \beta(c_i(s, a, s') + \gamma f((s', a'); \theta_k^i) - f((s, a); \theta_k^i))\nabla_\theta f((s, a); \theta_k^i), \tag{8}$$

$$\theta_{k+1}^i = \arg\min_{\theta \in \mathbf{B}} \left\| \theta - \theta_{k+1/2}^i \right\|_2, \tag{9}$$

where $s \sim \mu_{\pi_W}$, $a \sim \pi_W(\cdot|s)$, $s' \sim \mathsf{P}(\cdot|s, a)$, $a' \sim \pi_W(\cdot|s')$, $\beta$ is the learning rate, and $\mathbf{B}$ is a compact space defined as $\mathbf{B} = \{\theta \in \mathbb{R}^{md} : \|\theta - \theta_0^i\|_2 \leq R\}$. For simplicity, we denote the state-action pair as $x = (s, a)$ and $x' = (s'.a')$ in the sequel. We define the temporal difference error as $\delta_k(x, x', \theta_k^i) = f(x'_k, \theta_k^i) - \gamma f(x_k, \theta_k^i) - c_i(x_k, x'_k)$, stochastic semi-gradient as $g_k(\theta_k^i) = \delta_k(x_k, x'_k.\theta_k^i)\nabla_\theta f(x_k, \theta_k^i)$, and full semi-gradient as $\bar{g}_k(\theta_k^i) = \mathbb{E}_{\mu_{\pi_W}}[\delta_k(x, x', \theta_k^i)\nabla_\theta f(x, \theta_k^i)]$. We then describe the following regularity conditions on the stationary distribution $\mu_{\pi_W}$ and state-action value function $Q_{\pi_W}^i$, which are also adopted in the analysis of neural TD learning in Cai et al. (2019).

**Assumption 1.** *There exists a constant $C_0 > 0$ such that for any $\tau \geq 0$, $x \in \mathbb{R}^d$ with $\|x\|_2 = 1$ and $\pi_W$, it holds that $\mathbf{P}\left(|x^\top \psi(s, a)| \leq \tau\right) \leq C_0 \cdot \tau$, where $(s, a) \sim \mu_{\pi_W}$.*

**Assumption 2.** *We define the following function class:*

$$\mathcal{F}_{R,\infty} = \left\{ f((s, a); \theta) = f((s, a); \theta_0) + \int \mathbb{1}(\theta^\top \psi(s, a) > 0) \cdot \lambda(\theta)^\top \psi(s, a) dp(\theta) \right\}$$

*where $f((s, a); \theta_0)$ is the two-layer neural network corresponding to the initial parameter $\theta_0 = W_0$, $\lambda(\theta) : \mathbb{R}^d \to \mathbb{R}^d$ is a weighted function satisfying $\|\lambda(w)\|_\infty \leq R/\sqrt{d}$, and $p(\cdot) : \mathbb{R}^d \to \mathbb{R}$ is the density $D_w$. We assume that $Q_{\pi_W}^i \in \mathcal{F}_{R,\infty}$ for all $\pi_W$ and $i = \{0, \cdots, p\}$.*

**Assumption 3.** *For the visitation distribution of the global optimal policy $\nu^*$, there exist a constants $C_{RN}$ such that for all $\pi_W$, the following holds*

$$\int_x \left( \frac{d\nu^*(x)}{d\mu_{\pi_W}(x)} \right)^2 d\mu_{\pi_W}(x) \leq C_{RN}^2. \tag{10}$$

Assumption 1 implies that the distribution of $\psi(s,a)$ has a uniformly upper bounded probability density over the unit sphere. Assumption 2 is a mild regularity condition on $Q_{\pi_W}^i$, as $\mathcal{F}_{R,\infty}$ is a function class of neural networks with infinite width, which captures a sufficiently general family of functions. We further make the following variance bound assumption of neural TD update.

**Assumption 4.** *For any parameterized policy $\pi_W$ there exist a constant $C_\zeta > 0$ such that* $\mathbb{E}_{\mu_{\pi_W}} \left[ \exp \left( \left\| \bar{g}_k(\theta_k^i) - g_k(\theta_k^i) \right\|_2^2 / C_\zeta^2 \right) \right] \leq 1$ *for all $k \geq 0$.*

Assumption 4 implies that the expectation of variance error $\left\| \zeta_k(\theta_k^i) \right\|_2^2$ is bounded, which has been verified in (Cai et al., 2019, Lemma 4.5). The following lemma provides the convergence rate of neural TD learning. Note that the convergence rate of neural TD *in expectation* has already been establish in Cai et al. (2019); Wang et al. (2019). Here we characterize a stronger result on the convergence rate *in high probability*, which is needed for the analysis of our algorithm.

**Lemma 2** (Convergence rate of neural TD in high probability). *Considering the neural TD iteration defined in eq. (8). Let $\bar{\theta}_K = \frac{1}{K} \sum_{k=0}^{K-1} \theta_k$ be the average of the output from $k = 0$ to $K-1$. Let $\bar{Q}_t^i(s,a) = f((s,a), \theta_{K_{in}}^i)$ be an estimator of $Q_{\pi_{\tau_t W_t}}^i(s,a)$. Suppose Assumptions 1-4 hold, assume that the stationary distribution $\mu_{\pi_W}$ is not degenerate for all $W \in \mathbf{B}$, and let the stepsize $\beta = \min\{1/\sqrt{K}, (1-\gamma)/12\}$. Then, with probability at least $1 - \delta$, we have*

$$\left\| \bar{Q}_t^i(s,a) - Q_{\pi_{\tau_t W_t}}^i(s,a) \right\|_{\mu_\pi}^2 \leq \Theta \left( \frac{1}{(1-\gamma)^2 \sqrt{K}} \sqrt{\log \left( \frac{1}{\delta} \right)} \right) + \Theta \left( \frac{1}{(1-\gamma)^3 m^{1/4}} \sqrt{\log \left( \frac{K}{\delta} \right)} \right).$$

Lemma 1 implies that after performing the neural TD learning in eq. (8)-eq. (9) for $\Theta(\sqrt{m})$ iterations, we can obtain an approximation $\bar{Q}_t^i$ such that $\|\bar{Q}_t^i - Q_{\pi_{\tau_t W_t}}^i\|_{\mu_\pi} = \mathcal{O}(1/m^{1/8})$ with high probability.

**Constraint Estimation:** Since the state space is usually very large or even infinite in the function approximation setting, we cannot include all state-action pairs to estimate the constraints as for the tabular setting. Instead, we sample a batch of state-action pairs $(s_j, a_j) \in \mathcal{B}_t$ from the distribution $\xi(\cdot)\pi_{W_t}(\cdot|\cdot)$, and let the weight factor be $\rho_j = 1/|\mathcal{B}_t|$ for all $j$. In this case, the estimation error of the constrains $\left| \bar{J}_i(\theta_t^i) - J_i(w_t) \right|$ is small when the policy evaluation $\bar{Q}_t^i$ is accurate and the batch size $|\mathcal{B}_t|$ is large. We assume the following concentration property for the sampling process in the constraint estimation step, which has also been taken in Lan & Zhou (2016).

**Assumption 5.** *For any parameterized policy $\pi_W$ there exists a constant $C_f > 0$ such that* $\mathbb{E}_{\xi \cdot \pi_W} \left[ \exp([\bar{Q}_t^i(s,a) - \mathbb{E}_{\xi \cdot \mu_{\pi_{\tau_t W_t}}}[\bar{Q}_t^i(s,a)]^2 / C_f^2) \right] \leq 1$ *for all $k \geq 0$.*

**Policy Optimization:** In the neural softmax approximation setting, at each iteration $t$, an approximation of the natural policy gradient can be obtained by solving the following linear regression problem Wang et al. (2019); Agarwal et al. (2019):

$$\Delta_i(W_t) \approx \bar{\Delta}_t = \arg\min_{\theta \in \mathbf{B}} \mathbb{E}_{\nu_{\pi_{\tau_t W_t}}} [(\bar{Q}_t^i(s,a) - \phi_{W_t}(s,a)^\top \theta)^2]. \tag{11}$$

Given the approximated natural policy gradient $\bar{\Delta}_t$, the policy update takes the form of

$$\tau_{t+1} = \tau_t + \alpha, \ \ \tau_{t+1} \cdot w_{t+1} = \tau_t \cdot w_t + \alpha \bar{\Delta}_t \text{ (line 7) or } \tau_{t+1} \cdot w_{t+1} = \tau_t \cdot w_t - \alpha \bar{\Delta}_t \text{ (line 10)}. \tag{12}$$

Note that in eq. (12) we also update the temperature parameter by $\tau_{t+1} = \tau_t + \alpha$ simultaneously, which ensures $w_t \in \mathbf{B}$ for all $t$. The following theorem characterizes the convergence rate of Algorithm 1 in terms of both the objective function and constraint error.

**Theorem 2.** *Consider Algorithm 1 in the function approximation setting with neural softmax policy parameterization defined in eq. (7). Suppose Assumptions 1-5 hold. Suppose the same setting of policy evaluation step stated in Lemma 2 holds, and consider performing the neural TD in eq. (8) and eq. (9) with $K_{in} = \Theta((1-\gamma)^2 \sqrt{m})$ at each iteration. Let the tolerance $\eta = \Theta(m(1-\gamma)^{-1}/\sqrt{T} +$*

$(1 - \gamma)^{-2.5}m^{-1/8})$ *and perform the NPG update defined in eq.* (12) *with* $\alpha = \Theta(1/\sqrt{T})$. *Then with probability at least* $1 - \delta$, *we have*

$$J_0(\pi^*) - \mathbb{E}[J_0(\pi_{\tau_{out}W_{out}})] \leq \Theta\left(\frac{m}{(1-\gamma)\sqrt{T}}\right) + \Theta\left(\frac{1}{(1-\gamma)^{2.5}m^{1/8}}\log^{\frac{1}{4}}\left(\frac{(1-\gamma)^2T\sqrt{m}}{\delta}\right)\right),$$

*and for all* $i = 1, \cdots, p$, *we have*

$$\mathbb{E}[J_i(\pi_{\tau_{out}W_{out}})] - d_i \leq \Theta\left(\frac{m}{(1-\gamma)\sqrt{T}}\right) + \Theta\left(\frac{1}{(1-\gamma)^{2.5}m^{1/8}}\log^{\frac{1}{4}}\left(\frac{(1-\gamma)^2T\sqrt{m}}{\delta}\right)\right).$$

*where the expectation is taken only with respect to the randomness of selecting* $W_{out}$ *from* $\mathcal{N}_0$.

Theorem 2 guarantees that CRPO converges to the global optimal policy $\pi^*$ in the feasible set at a sublinear rate $\mathcal{O}(1/\sqrt{T})$ with an optimality gap $\mathcal{O}(m^{-1/8})$, which vanishes as the network width $m$ increases. The constraint error bound of the output policy converges to zero also at a sublinear rate $\mathcal{O}(1/\sqrt{T})$ with a vanishing optimality gap $\mathcal{O}(m^{-1/8})$ as $m$ increases. The optimality gap arises from both the policy evaluation and policy optimization due to the limited expressive power of neural networks. To attain a $w_{out}$ that satisfies $J_0(\pi^*) - \mathbb{E}[J_0(\pi_{\tau_{out}W_{out}})] \leq \epsilon + \Theta(m^{-1/8})$ and $\mathbb{E}[J_i(\pi_{\tau_{out}W_{out}})] - d_i \leq \epsilon + \Theta(m^{-1/8})$, CRPO needs at most $T = \mathcal{O}(m^2\epsilon^{-2})$ iterations, with each iteration contains $\Theta(\sqrt{m})$ policy evaluation iterations. The convergence analysis in the function approximation setting is more challenging than that in the tabular setting. Since the class of neural softmax policy is not complete, we need to handle additional approximation errors introduced by the neural network parameterization. It is worth noting that CRPO is the first SRL algorithm that has global optimal guarantee in the function approximation setting over general CMDP.

## 5 EXPERIMENT

We conduct experiments based on OpenAI gym Brockman et al. (2016) that are motivated by SRL. We consider two tasks with each having multiple constraints given as follows:

- **Cartpole:** The agent is rewarded for keeping the pole upright, but is penalized with cost if (1) entering into some specific areas, or (2) having the angle of pole being large.
- **Acrobot:** The agent is rewarded for swing the end-effector at a specific height, but is penalized with cost if (1) applying torque on the joint when the first link swings in a prohibited direction, or (2) when the the second link swings in a prohibited direction with respect to the first link.

The detailed experimental setting is described in Appendix A. For both experiments, we use neural softmax policy with two hidden layers of size $(128, 128)$. In previous studies, PDO and CPO have been widely adopted as baseline algorithms. Since we are considering multiple constraints and do not assume the accessibility of a feasible policy as an initialization, baseline algorithm CPO is not applicable here. Thus, we compare CRPO only with PDO in our experiments. For fair comparison, we adopt TRPO as the optimizer for both CRPO and PDO. In PDO, we initialize the Lagrange multiplier as zero in both tasks. The learning curves for CRPO and PDO are provided in Figure 1. At each step we evaluate the performance based on two metrics: the return reward and constraint value of the output policy. We show the learning curve of unconstrained TRPO (the green line), which although achieves the best reward, but does not satisfy the constraints, i.e., the optimal policy obtained by such an unconstrained method is infeasible. In both tasks, CRPO tracks the constraints

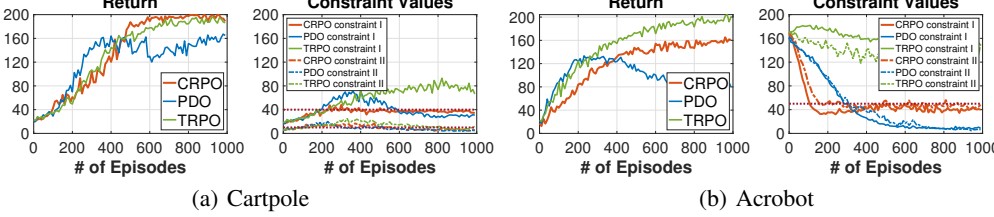

|  (a) Cartpole | (b) Acrobot |

Figure 1: *Average performance for CRPO, PDO, and unconstrained TRPO over 10 seeds. The red dot lines in (a) and (b) represent the limits. In Cartpole, the limits of two constraints are 40 and 10, respectively. In Acrobot, the limits of both constraints are 50.*

return almost exactly to the limit, indicating that CRPO sufficiently explores the boundary of the feasible set, which results in an optimal return reward. In contrast, although PDO also outputs a constraints-satisfying policy in the end, it tends to over- or under-enforce the constraints, which results in lower return reward and unstable constraint satisfaction performance.

## 6 Conclusion

In this paper, we propose a novel CRPO approach for policy optimization in the CMDP setting, which is easy to implement and has provable global optimality guarantee. We show that CRPO achieves an $\mathcal{O}(1/\sqrt{T})$ convergence rate to the global optimum and an $\mathcal{O}(1/\sqrt{T})$ rate of vanishing constraint error when NPG update is adopted as the optimizer. This is the first finite-time analysis for SRL algorithms under general CMDP. In the future, it is interesting to incorporate various momentum schemes to CRPO to improve its convergence performance.

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
