# OpenReview forum: "A Primal Approach to Constrained Policy Optimization: Global Optimality and Finite-Time Analysis"
_ICLR.cc/2021/Conference — Reject_

### Official Review · AnonReviewer4 · 2020-10-27
**good theory paper**

**Rating:** 7
**Confidence:** 4

**Review:**

This paper considers a primal approach to the constrained RL problem where the constraints have a similar form as the total reward. The paper establishes global convergence in the tabular and NTK approximation cases. The problem in the aforementioned two cases is non-convex and therefore the global convergence results are very interesting and can contribute to the RL theory community. I did not check the entire proof but believe it is correct after checking some key points and go through the technical lemmas. The assumption on the one hidden layer neural network is standard, as it is used in a series of recent literature, although it is strong compared with practical algorithms.

I have a question about Lemma 1, which is borrowed from another paper and I don't have enough time to check the specific lemma in that paper and its corresponding contextures and proof. The result says policy evaluation converges to the true Q function under a uniform measure (i.e. 2-norm), while the evaluation sample comes from the policies during iteration, which probably chooses sparse actions and visits sparse states. Do I miss any assumption on the exploration ability of the policies during iteration? Similar questions arise in applying Lemma 2 to Theorem 2. In the papers on unconstrained MDP (Agarwal 19 and Wang 19), they assume bounds on the ratio of visitation measures between policies to handle this technique problem. While I seem to can not find the corresponding parts and would like to ask about the intuition behind that.

---

> ### Author Response · Authors · 2020-11-17
> **We thank the reviewer very much for the very helpful comments! Below is our response**
>
> Thank you so much for pointing out these. In fact we also make similar assumptions as in (Agarwal 19 and Wang 19) to bound the ratio of visitation measures between policies (see eq (21) in the appendix) but we did not state this assumption in the main text. We will move this assumption from appendix to our main text in our updated version.

---

### Official Review · AnonReviewer1 · 2020-10-28
**Comments**

**Rating:** 5
**Confidence:** 3

**Review:**

This paper proposes an algorithm called CRPO to solve the constraint reinforcement learning. Different from the literature, where the primal-dual formulations are widely used, this paper introduces a pure primal algorithm. The motivation of this paper is clear, since in practice the primal-dual formulation is hard to train and needs much effort on the hyperparameter tuning. Thus I think the pure primal algorithm is an interesting direction of constrained RL. Another merit of this paper is that it avoids the projection step in literature. It does the gradient step on the constraint when it is not satisfied. The author provides the convergence result both on the tabular setting and function approximation setting (two layer neural network), where it has already been known that the natural gradient can converge to the global optimal (tabular setting ) and the (shallow) neural network behaves like a convex function. I did not check the details of the proof, but the  roadmap of the proof is clear by combining existing results on RL and deep learning. At last, the author affirms their theoretical result by two toy examples on cartpole and acrobot.

My main concern comes from the novelty on the theoretical result, particularly the proof technique.  The result on the policy evaluation step mainly comes from Dalal et al, while the policy optimization step stems from Agarwal et al. When combined with function approximation setting, it adopts the analysis of the neural networks such as the result in  Du et al. The whole framework is similar to the Lan & Zhou. It seems that the author just combines all these existing results together. Can you emphasize the main theoretical contribution of this paper?

Although this work avoids the primal-dual formulation or projection step, my feeling is that it still needs careful hyperparameter tuning. For instance , how do you pick the value of eta in the algorithm? How many policy evaluations are needed before the policy update step?

In section 4.2. The result is on the two-layer neural network(one hidden layer). However, the algorithm is tested on two hidden layer (with size (128,128)) neural networks. Is it possible to extend the theoretical result to multi-layer (more than one hidden layer) neural network?

---

> ### Author Response · Authors · 2020-11-17
> **We thank the reviewer very much for the very helpful comments! Below is a point-to-point response**
>
> Q1: Emphasize the novelty of the theoretical result.
>
> A1: First of all, our main contribution lies in proposing a new and easily implementable algorithm to the safety RL community, which can be potentially very useful for safety RL applications. Our theory serves as a supporting role to show that rather than a heuristic algorithm, our proposed algorithm is the first primal constrained policy optimization algorithm that has global optimal guarantee in multi-constraint setting.
> Regarding the novelty of theory development, our analysis goes significantly further beyond combining the existing pieces of proofs. The following are two nontrivial developments in our proof that are not captured in the previous works:
> 1. The algorithm in Lan & Zhou only considers single constraint settings. We develop a new framework to show that CRPO is applicable to the multi-constraint setting. This development is not straightforward.
>
> 2. We explain below that new challenges do arise beyond Lan & Zhou and Agarwal et al, which requires a new analysis framework.
>
> (1) CRPO adopts the actor-critic structure, and uses critics to estimate the constraints and help the actor to estimate the policy update, while in the CSA algorithm (Lan & Zhou), unbiased estimators of constraints and convex gradient are both accessible. Thus, analyzing CRPO is more challenging than analyzing CSA in (Lan & Zhou).
>
> (2) Moreover, since the randomness of the interaction error between the actor and critic affects how the algorithm switches between objective and constraints, technique in NPG (Agarwal et al.) is not applicable to capture the overall convergence performance due to the dynamically changing optimization objective. Due to these challenges, simply combining the technique in (Lan & Zhou) and (Agarwal et al.) is not sufficient to establish the convergence proof of CRPO.
>
> To handle the issues in (1) and (2), we develop a novel analysis framework, in which we focus on the event in which the critic returns almost accurate value function estimation, and establishes the convergence of CRPO in this event. In such an event, we can better capture how the algorithm switches between objective and multiple constraints so that we can develop the convergence rate of CRPO in this more challenging setting.
>
> Q2: Whether carefully parameter tuning is necessary for CRPO?
>
> A2: (1) For $\eta$: In both experiments, we let $\eta = 0.5$. In fact, we find that the performance of CRPO is not very sensitive to the value of $\eta$ over a wide range, i.e, letting $\eta$ from 0.05 to 2 for CRPO yields almost the same convergence performance. However, in primal-dual optimization (PDO), the convergence performance is very sensitive to the stepsize of the dual variable. Recall that in our experiment we adopt 0.0005 as the best tuned dual variable stepsize. We find that the performance of PDO with dual variable stepsize 0.0001 and 0.001 is significantly worse than that of PDO with dual stepsize 0.0005. In order to tune PDO to have a good performance, we need to be really careful with the additional parameters introduced by the dual variable. Thus, in contrast to the difficulty of tuning the PDO algorithm, CRPO is much less sensitive to hyper-parameters and is hence much easier to tune.
>
> (2) For policy evaluations step: In practice we let the critic inherit parameters from the last iteration (which is widely adopted in standard AC-type algorithms -- see open source code in OpenAI spinning up). In both experiments, we find that the performance of our CRPO is not sensitive to the policy evaluation steps, i.e, selecting policy evaluation steps from 1 to 20 at each iteration yields almost the same overall convergence performance. In our experiment, we found that updating the critic with one step from the last iteration is already good enough to guarantee a good empirical performance. Thus, selecting the policy evaluation step does not cause too much tuning effort in performing the CRPO algorithm.
>
> Q3: Can the result be extend to DNN?
>
> A3: Yes, it is possible to extend our result to DNN by adopting similar steps in a recent paper.
> Fu, Z., Yang, Z., & Wang, Z. (2020). Single-timescale actor-critic provably finds globally optimal policy. arXiv preprint arXiv:2008.00483. (see the proof in Section 3.2)

---

### Official Review · AnonReviewer3 · 2020-10-28
**simple & working algorithm, but possibly incomplete theory**

**Rating:** 6
**Confidence:** 3

**Review:**

This paper considers safe RL through solving a constrained MDP in (1). The authors proposed a primal method which alternates between maximizing the reward and minimizing the constraint violation. The convergence rate of the algorithm is also provided under standard settings, e.g., bounded reward, iid samples, etc.. Interestingly, the authors also analyzed the case for using a 2-layer neural network (NN) function approximator for the policy and the actor.

On the upside, this paper suggested a simple algorithm for a new problem that has not been studied extensively yet and a number of settings have been covered. The authors also supplemented this work with several numerical experiments. The reviewer has a few comments / suggestions regarding the NN function approximator setting as follows:

- Convergence results of Theorem 2

While the convergence is well analyzed for the tabular setting in Theorem 1, the reviewer is concerned about the result in Theorem 2 for the case analyzed with function approximator. The latter shows that the optimality gap decays with $T,m$ as $O(m/\sqrt{T} + \log(T)/m^{1/8})$ - which indicates that the iteration complexity grows with $m$, i.e., the width of the 2-layer NN. This seems to be non-ideal compared to many recent works on 2-layer NN, e.g., (Cai et al., 2019) as cited in this paper, where the iteration complexity is often independent of $m$. Is there any reason behind such discrepancy in the rate? Or is there any numerical evidence which shows that such dependence on $m$ is unavoidable?

The reviewer suspects that the result in Theorem is not tight, which can be possibly improved.

- Implementation of proposed algorithm with function approximator

Algorithm 1 combines a policy evaluation step and the policy gradient step. In particular, the policy evaluation needs to be run with multiple steps to obtain a sufficiently accurate solution. In the case of using NN as function approximator, Lemma 1 for neural TD (Cai et al., 2019) holds with high probability w.r.t. the NN initialization, and the analysis in this paper suggests that the NN needs to be reinitialized every time when the policy is changed. Is it the same way that the algorithm is initialized?

On the other hand, the reviewer also wonders if such "restarting" is necessary given that the policy is updated with only 1 step of policy gradient.

* Minor point: It was mentioned that the CSA method in Lan and Zhou (2016) handles only convex functional constraints which is different from the current work. However, it is worth pointing out that in the tabular setting, the constraint, objective functions are indeed close to a linear function with respect to the policy, i.e., it is roughly convex. The authors may shed more lights on the similarity between the proposed algorithm and (Lan and Zhou, 2016).

* Minor point: in the statement of Lemma 6 & 7, since the results are specialized for the case of $i=0$. It is better to use $i=0$ in the equation.

* Minor point: in Assumption 1, it is not clear what do the authors mean by $P( |x^T \psi(s,a)| ) \leq C_0 \tau$. What is this event of $|x^T \psi(s,a)|$?

===== Post Rebuttal =====
The reviewer is satisfied by the authors' response. I am fine with raising my score. In the final paper, it would be nice if the authors can include a detailed discussion about the dependence of $m$ in their results.

---

> ### Author Response · Authors · 2020-11-17
> **We thank the reviewer very much for the very helpful comments! Below is a point-to-point response**
>
> Q1: Why convergence rate have dependence on $m$? Is the bound tight?
>
> A1: The sample complexity results in (Cai et al., 2019, Wang et al. 2020) are also dependent on the width of the 2-layer neural network (see Theorem 4.6 & 4.7 in Cai et al. 2019). The reason that the “width” sometimes does not show up in their error term is because they let the width $m$ to be dependent on the iteration number $T$ (e.g., see Corollary 4.14 in Wang et al. 2020), so that their complexity result is expressed only with the dependence on $T$.
>
> In fact, high-level speaking, the optimality gap must depend on the width of the neural network in general, because the approximation of the true Q function by a neural network necessarily causes an error that depends on the expensive power of the neural network. As $m$ increases, the expressive power of the neural network increases, and hence the approximate error decreases.
>
> Moreover, it is not fair to compare our result of CMDP with the convergence result of NPG in unconstrained MDP (Wang et al. 2020, Cai et at. 2019). In CMDP, we need to handle the nonsmoothness of the objective function introduced by the constraints, i.e., in CMDP, we consider to optimize the function $J_0(w)1_{J_1(w)\leq d + \eta} + J_1(w)1_{J_1(w)> d + \eta}$, which is highly nonsmooth, while in unconstrained MDP, we only consider to optimize the function $J_0(w)$, which is smooth. Thus the objective function in CMDP (our paper) is more difficult to optimize than the objective function considered in constrained MDP (Wang et al. 2020, Cai et al. 2019). Generally speaking, in nonsmooth problems, both the Lipschitz conditional numbuer and the variance are larger than those in smooth problems. Since both the Lipschitz conditional number and variance are depend on $m$ in 2-layer NN model, it is not suprising that the convergence result of smooth problem in (Wang et al. 2020, Cai el at. 2019) may have better dependence on $m$ than the result of nonsmooth problem provided in our paper.
>
> As far as we can see, our bound matches the existing state-of-the-art result (Wang et al. 2020, Cai et al. 2019) when the setting reduces to unconstrained problems.
>
> Q2: Is it the same way that the algorithm is initialized? Is "restarting" necessary?
>
> A2: We want to clarify that our analysis and theory still hold if the critic (policy evaluation part) inherits the parameter from the last iteration, which is exactly what we did in our experiment. In our experiment, we actually implement the algorithm without reinitialization, and let the critic inherit the parameter from the last iteration. In practice, one step for each critic’s update is typically good enough for the convergence, but in theory we require the critic to update several steps to achieve a certain accuracy in order to guarantee the convergence in the worst case.
>
> Minor point:
> 1. Thanks for the suggestion, we will add more comparison in our paper. In fact, our analysis is very different from that in Lan & Zhou in the following senses.
> (1) Lan & Zhou only consider single constraint setting, while our CRPO is capable to handle multi-constraints setting, thus is more challenging to analyze
> (2) Lan & Zhou assume the accessibility of unbiased estimators of both gradient and constraint, while in our problem both the NPG update and constraints are estimated through the random output from the critic, thus requiring developing a new analysis framework to handle this more challenging setting.
> 2. Thanks for pointing out this, we will modify that in our appendix.
> 3. Sorry for the confusion, this is a typo. The correct form should be $P( |x^T \psi(s,a)| \leq \tau ) \leq C_0 \tau$. We will correct that in our updated version. Thank you for pointing out this typo.

---

### Official Review · AnonReviewer2 · 2020-11-03
**Review for "A Primal Approach to Constrained Policy Optimization: Global Optimality and Finite-Time Analysis"**

**Rating:** 5
**Confidence:** 2

**Review:**

This paper proposes a new method for constrained MDP. The proposed method does not require primal-dual formulation and is easy to implement given the availability of state-of-the-art policy optimization solvers.  When the natural policy gradient is used as the  policy optimizer,  a sublinear global convergence rate is proved for both the tabular setting and the function approximation case.



Pros:
1.  This paper is quite original. Although the paper is inspired by the cooperative stochastic approximation method in optimization literature, significant efforts have been made to adapt things to the CMDP setup.

2. This paper claims to have some global convergence results for CMDP. This is significant.

3. The proposed method is easy to implement. The constrained policy optimization problem is almost transferred to some unconstrained sub-tasks such that existing policy optimization solvers can be directly used.

Cons:

1. I don't follow the proof very well. I am not convinced how the proposed method achieved global optimality. Even the high-level proof idea is unclear to me. Here is one example confusing me. Say we t have two functions J_1 and J_0. In addition, we have J_1=J_0. It seems possible that we will have some limit cycle in this case. One does a maximization on J_0 but realizes the constraint on J_1 is violated. Then a minimization on J^1 is done and then the next step of maximizing J^0 may send the policy weight back to the same point.  On the conceptual level, minimizing the constraint function can send the policy weight towards the opposite direction of maximizing J_0, and the iterations may bounce around and be stuck in limit cycle (around local min). Can the authors provide some proof sketch to explain why such cases can be avoided?

2. I am also confused by the assumptions. Does the author assume that there exists a unique solution for the CMDP problem? Can the authors clarify the assumptions in Section 4.2? Are these assumption true for the two examples considered in this paper?

3. The numerical examples need some improvements. It will be more convincing if the authors can demonstrate their methods on
other standard benchmarks for constrained policy optimization (e.g. Gather, Circle, Half-Cheetah Safe, etc).

---

> ### Author Response · Authors · 2020-11-17
> **We thank the reviewer very much for the very helpful comments! Below is a point-to-point response.**
>
> Q1: Proof sketch of the "limit-cycle" example.
>
> A1: High-level speaking, once the oscillation occurs, the points are already in the neighbourhood of the global/local minimum because a single update to move the reward up would violate the constraint. Second, the range of the oscillation depends on the stepsize, which is set to be $\alpha = \Theta(1/\sqrt{T})$. By setting $T$ large enough, the required convergence accuracy can be satisfied. To argue the limit is global rather than local minimum, in fact it is well known  (e.g., Agarwal et al. 2019,Wang et al. 2019) that NPG by nature updates along the descent direction in the probability (i.e., policy) space, and hence is guaranteed to move towards the direction of the global optimum of the objective function in RL. Hence, since CRPO applies NPG to either the objective function or the constraint function, it is not attracted at local minimum.
>
> We next provide a mathematical proof sketch to show that if the “limit-cycle” happens between two points $w_1$ and $w_2$, then our CRPO would select one of the two points, say $w_1$, as the final output, and $\pi_{w_1}$ is an almost-feasible, almost-optimal solution satisfies the following two conditions:
> 1. $\pi_{w_1}$ is outside the feasible set but very close to the feasible set boundary, i.e., $J(w_1) \leq d + \eta$.
> 2. $\pi_{w_1}$ has larger function value than the global optimal policy in the feasible set, i.e., $J(w_1)>J(\pi^*)$.
>
> We define $w_1$ as the point when objective function is updated ($J(w_1) \leq d + \eta$), and $w_2$ as the point when constraint is updated ($J(w_2) > d + \eta$), i.e., $w_2 = w_1 + \alpha \text{NPGupdate}(w_1)$ and $w_1 = w_2 - \alpha\text{NPGupdate}(w_2)$. Without loss of generality, we assume the algorithm is initialized at $w_1$.
>
> Step 1: Following NPG update rule, we have the following holds: $\alpha(J(\pi^*) - J(w_1)) \leq D(\pi^*| \pi_{w_1}) -  D(\pi^*| \pi_{w_2}) + \alpha^2 C$  (1) and $\alpha(J(w_2) - J(\pi^*)) \leq D(\pi^*| \pi_{w_2}) -  D(\pi^*| \pi_{w_1}) + \alpha^2 C$ (2), where $C$ is a constant.
>
> Step 2: Summing (1) and (2) and dividing both sides with $\alpha$ yield $ (J(\pi^*) - J(w_1)) + (J(w_2) - J(\pi^*)) \leq 2\alpha C $. Note that $J(w_2) \geq d + \eta$ (due to the update rule see line 9 in Algorithm 1) and $d \geq J(\pi^*)$ (because $d$ is the constraint limit that is larger than all the value in the feasible set). We can further obtain $ (J(\pi^*) - J(w_1)) + \eta \leq 2\alpha C $ (3).
>
> Step 3: If we let $\eta = 2\alpha C$, then (3) implies $J(w_1)\geq J(\pi^*)$ ($w_1$ is outside the feasible set in this case). Recall that our algorithm selects output from iterations in which the objective function is optimized, and our algorithm oscillates between $w_1$ and $w_2$  throughout the entire iterations in this case, thus $w_1$ is the final output of our algorithm. Since $J(w_1) \leq d + \eta$, $w_1$ is very close to the boundary of the feasible set.
>
>
> Q2: Does the author assume a unique solution for CMDP? Clarify assumptions in 4.2? Is the assumption true for two examples?
>
> A2: (1) We did not assume a unique solution for CMDP. There can be multiple global optimum in the feasible set and those global optimum share the same function value $J(\pi^*)$. Our CRPO algorithm will converge to one of them.
>
> (2) Assumption 1 can be satisfied for most of the ergodic Markov chain. For simplicity, without loss of generality we can let $||\psi(s,a)||_2 = 1$. In this case, Assumption 1 holds when the stationary distribution of the MDP has bounded distribution density, which is a weak assumption.
>
> Then, we want to clarify that Theorem 2 still holds without Assumption 2, but with an additional approximation error $\epsilon$ added to the right hand side of the convergence result ($\epsilon$ indicates the error of approximating Q-function using a neural network). Here we include Assumption 2 for theoretical convenience so that we can capture the relationship between $\epsilon$ and the number of neurons $m$.
>
> We also want to clarify that Theorem 2 still holds without Assumptions 3 & 4, but with slightly different dependence on the factor $1/\delta$. In fact, Assumptions 3 & 4 are standard in the convergence proof with high probability (see Ghadimi et al 2013). Here we include Assumptions 3 & 4 for theoretical convenience so that Theorem 2 has logarithmic dependence on $1/\delta$. If we remove Assumptions 3 & 4, the convergence result in Theorem 2 will have polynomial dependence on $1/\delta$. Assumptions 3 & 4 depend on the nature of the MDP, and hold true if the stationary distribution of the MDP satisfies “light-tail” property.
>
> Q3: The numerical examples need some improvements.
>
> A3: Thank you for the suggestion. We are currently working on the suggested experiments. If we obtain any result before the rebuttal deadline, we will update our paper and let you know.

---

### Decision · Program_Chairs · 2021-01-07
**Final Decision**

**Decision:**

Reject

**Comment:**

All reviewers appreciated the main result in the paper, which gives  global optimality guarantee for constrained policy optimization for both tabular setting and NTK setting. However, there were a number of unclear parts of the paper reported by several reviewers (assumptions, hyperparameter tuning, complexity dependence on the number of neurons, experimental setups). On top of it, the AC also echoes with R1’s concern about the novelty of this work as it basically stacks existing results (TD by Dalal et al., Neural TD by Cai et al. (2019), NPG by Agarwal et al, CSA algorithm by Lan & Zhou).
These concerns made me reticent to recommend acceptance at this point. I strongly encourage the authors to continue their interesting work in considering the reviewer comments and strengthen the numerical experiments.